# Combining Diffusion, Convection and Absorption: A Pilot Study of Polymethylmethacrylate versus Polysulfone Membranes in the Removal of P-Cresyl Sulfate by Postdilution On-Line Hemodiafiltration

Pablo Molina [1,2,*], Julio Peiró [3], María A. Martínez-Gómez [4], Belén Vizcaíno [1], Cristina Esteller [3], Mercedes González-Moya [1], María García-Valdelvira [3], Mariola D. Molina [5], Francisco Maduell [6] and on behalf of the Collaborators [†]

1. Department of Nephrology, Hospital Universitari Dr. Peset. FISABIO, 46017 Valencia, Spain; belvizcaino@gmail.com (B.V.); mgonzalezm@alumni.unav.es (M.G.-M.)
2. Department of Medicine, Universitat de València, 46010 Valencia, Spain
3. Department of Clinical Analysis, Hospital Universitari Dr. Peset. FISABIO, 46017 Valencia, Spain; julio.peirog@gmail.com (J.P.); cristina_ebxert@hotmail.com (C.E.); maryagv@msn.com (M.G.-V.)
4. Department of Pharmacy, Hospital Universitari Dr. Peset. FISABIO, 46017 Valencia, Spain; martinez_margoma@gva.es
5. Department of Mathematics, Universidad de Alicante, 03690 Alicante, Spain; mariola.molina@ua.es
6. Department of Nephrology, Hospital Clínic, 08036 Barcelona, Spain; fmaduell@clinic.cat
* Correspondence: molina_pab@gva.es
† The investigators who participated in the study are listed in Appendix A.

**Abstract:** Dialytic clearance of p-cresyl sulfate (pCS) and other protein-bound toxins is limited by diffusive and convective therapies, and only a few studies have examined how to improve their removal by adsorptive membranes. This study tested the hypothesis that high-flux polymethyl-methacrylate (PMMA) dialysis membranes with adsorptive capacity increase pCS removal compared to polysulfone membranes, in a postdilution on-line hemodiafiltration (OL-HDF) session. Thirty-five stable hemodialysis patients randomly completed a single study of 4 h OL-HDF with PMMA (BG2.1U, Toray®, Tokyo, Japan) and polysulfone (TS2.1, Toray®) membranes. The primary endpoint was serum pCS reduction ratios (RRs) obtained with each dialyzer. Secondary outcomes included RRs of other solutes such as β2-microglobulin, the convective volume obtained after each dialysis session, and the dialysis dose estimated by ionic dialysance (Kt) and urea kinetics (Kt/V). The RRs for pCS were higher with the PMMA membrane than those obtained with polysulfone membrane (88.9% vs. 58.9%; $p < 0.001$), whereas the β2-microglobulin RRs (67.5% vs. 81.0%; $p < 0.001$), Kt (60.2 ± 8.7 vs. 65.5 ± 9.4 L; $p = 0.01$), Kt/V (1.9 ± 0.4 vs. 2.0 ± 0.5; $p = 0.03$), and the convection volume (18.8 ± 2.8 vs. 30.3 ± 7.8 L/session; $p < 0.001$) were significantly higher with polysulfone membrane. In conclusion, pCS removal by OL-HDF was superior with high-flux PMMA membranes, appearing to be a good dialysis strategy for improving dialytic clearance of pCS, enabling an acceptable clearance of β2-microglobulin and small solutes.

**Keywords:** adsorption; chronic hemodialysis; dialyzer membrane; hemodiafiltration; p-cresyl sulphate; polymethylmethacrylate; polysulfone; uremic toxins

## 1. Introduction

Chronic kidney disease (CKD) is characterized by the progressive accumulation of multiple chemical compounds that are normally excreted into the urine in healthy people [1–5]. These chemical compounds are globally known under the name of uremic toxins and are conventionally classified into three groups based on their physicochemical properties [2]. These major groups include small water-soluble compounds with molecular weight less than 500 Da, larger molecules with molecular weight greater than 500 Da

(principally small peptidic compounds), and protein-bound uremic toxins (PBUTs). Most PBUTs are small hydrophobic molecules, with p-cresyl sulfate (pCS), indoxyl sulfate (IS), and 3-carboxy-4-methyl-5-propyl-2-furanpropionic acid (CMPF) being the most widely studied [3–6].

pCS is an important representative of the PBUTs, which are linked to cardiovascular outcomes in CKD patients [7–10]. Efforts are mounting to reduce serum concentrations, either by reducing intestinal uptake of nutrients [11–13] or by improving blood clearances [14–18]. Dialytic clearance of pCS and other PBUTs is poor by diffusive treatment and limited by high-flux hemodialysis (HD) and on-line hemodiafiltration (OL-HDF), and only a few studies have examined how to improve their removal by other extracorporeal strategies [4,8–10,12].

Polymethylmethacrylate (PMMA) membrane was designed to offer high biocompatibility to patients and add adsorption to the known HD mechanisms of diffusion and convection. This membrane is characterized by a symmetric structure with large, long, and winding pores, providing a better hydrophobic and cationic adsorption capability than other synthetic membranes, such as polysulfone (PS) [19–22]. While diffusion and convection remove small and medium molecules, adsorption allows the removal of medium and high molecular weight molecules, including PBUTs, responsible for many complications in the uremic patient [23]. Due to their low permeability and high cut-off with high albumin loss, first generations of PMMA dialyzers (e.g., BK series) showed a less appropriate profile for use in HDF than PS. However, since they began in 1977, the PMMA membranes have been improved and adapted to new treatment modalities, such as OL-HDF [24]. BG-U is one of the latest PMMA series designed by Toray, remaining highly adsorptive but with high permeability capacity and a comparable cut-off to high-flux PS dialyzers [25].

As high-flux PS membranes with a restricted cut-off demonstrate low effectivity for the removal of PBUTs, we have studied whether the removal ratios of such uremic toxins may be improved by using adsorptive membranes such as PMMA. The aim of this study was to evaluate the effectiveness of a high-flux PMMA membrane (BG-2.1U) compared to a high-flux PS membrane (TS-2.1SL) for the removal of the PBUT pCS in patients undergoing postdilution OL-HDF.

## 2. Materials and Methods

### 2.1. Study Design and Patients

This prospective, single-center, cross-over study enrolled thirty-five adult HD patients who were stable on a thrice-weekly HD program (HD performed 4.0 to 4.5 h), for at least 3 mo, and agreed to give informed consent. All patients were anuric with urine volume of <100 mL/day. Patients were excluded in the case of any serious clinical situation that would lead to an expected survival of less than 1 year. The study was approved by the Dr Peset Hospital Research Ethics Committee (approval number: 14/012). Written informed consent was obtained for all participating patients, in accordance with the Declaration of Helsinki.

### 2.2. Hemodialysis Procedures

Each patient underwent two OL-HDF sessions with usual dialysis parameters: dialysis buffer with bicarbonate, dialysate flow rate (Qd) 500 mL/min, blood flow rate (Qb) between 350 to 450 mL/min, and dialysis time between 4.0 and 4.5 h. All patients received postdilution OL-HDF with automatic adjustment of the substitution fluid flow rate, to maximize substitution volume while simultaneously avoiding hemoconcentration and filter clotting [26]. All treatments were performed with the 5008 HD system (Fresenius Medical Care), and with ultrapure dialysis fluid, containing <0.1 colony-forming unit/mL and <0.03 endotoxin unit/mL. Treatment parameters, including blood and dialysate flow rates, length of the dialysis session, and ultrafiltration rate, remained unchanged during both sessions. The only difference among the two dialysis sessions in each patient was the dialyzer: high-flux PMMA BG2.1U (Toray®, Tokyo, Japan) and high-flux PS TS2.1SL

(Toray®, Tokyo, Japan). Differences and similarities of both membranes are shown in Table 1. All the sessions were performed in the intermediate period of Wednesday or Thursday, with a 4-week interval between study sessions. During this wash-out period, patients remained in their usual HD treatment plan with no changes; all received postdilution OL-HDF with high-flux, PS FX-100 (Fresenius Medical Care®, Bad Homburg vor der Höhe, Germany). The order of the two different treatment sessions was randomly assigned to the patients.

**Table 1.** Technical characteristics of the dialyzers. Adapted from Cavalier, 2017; Masakane, 2017; and Gómez, 2020 [25,27,28].

| Characteristic | BG-2.1U (Toray®) | TS-2.1SL (Toray®) |
|---|---|---|
| Surface area (m$^2$) | 2.1 | 2.1 |
| Membrane structure | PMMA | PS |
| Sterilization | γ radiation | γ radiation |
| Membrane thickness (μm) | 30 | 40 |
| Internal diameter (μm) | 200 | 200 |
| Membrane frame | Symmetrical | Asymmetrical |
| Pore diameter (Å) | 70 | 25 |
| Negative charge (mEq/g) [1] | 110 | NA |
| KUF in vitro (mL/h) [2] | 4300 | 5500 |
| SC β2-microglobulin | NA | 0.93 |
| SC myoglobin | NA | 0.7 |
| SC albumin | <0.05 | <0.003 |
| Urea clearance (mL/min) [3] | 192 | 199 |
| Creatinine clearance (mL/min) [3] | 191 | 197 |
| Phosphate clearance (mL/min) [3] | 179 | 196 |
| Vitamin B$_{12}$ clearance (mL/min) [3] | 133 | 171 |
| Inulin clearance (mL/min) [1] | 81 | 142 |

Abbreviations: KUF, ultrafiltration coefficient; NA, not available; PMMA, polymethylmethacrylate; PA, polysulfone; SC, sieving coefficient. [1] Estimated by the reaction of iodine and iodide ions after dried hollow fiber membrane immersed in a solution of 5% potassium iodide in methanol for 24 h. [2] Measured with bovine blood (hematocrit, $30 \pm 3\%$; total protein, $6.0 \pm 0.5$ g/dL; blood flow rate, $200 \pm 4$ mL/min; transmembrane pressure, $13.3 \pm 1.3$ kPa; temp., $37 \pm 1$ °C). [3] In vitro clearances with aqueous solution (Qb $200 \pm 4$ mL/min; Qd $500 \pm 10$ mL/min).

### 2.3. Study Outcomes and Measurements

The primary outcome measures were the percentage reduction ratio (RR) in pCS (188 Da) concentrations from pre- to posttreatment, using standard techniques [29]. The secondary outcomes considered RR of other solutes, including β$_2$-microglobulin (11,800 Da), urea (60 Da), phosphate (95 Da), creatinine (113 Da), and uric acid (168 Da), as well as the convective volume and the dialysis dose estimated by ionic dialysance (Kt) and urea kinetics (Kt/V), obtained after each dialysis session [30]. The pre- to posttreatment reduction ratios for pCS and β$_2$-microglobulin were determined after correcting postdialysis concentrations for the extracellular fluid contraction due to ultrafiltration, using the method of Bergström and Wehle (uncorrected postdialysis β2-microglobulin/[1 + (Δbody weight/0.2 postdialysis weight)]) [31].

### 2.4. Blood Sampling

Predialytic blood samples were collected after insertion of the access needle, ensuring that the blood was not diluted by saline or heparin. The postdialytic sample was drawn from the arterial needle after slowing the blood pump to 50 mL/min [29]. Blood samples were collected in serum gel tubes and then were left to stand for a minimum of 50 to 60 min before centrifuging for 10 min at 3500 rpm. Serum was then separated, analyzed for small water-soluble compounds and β2-microglobulin, and finally frozen at −80 °C until analysis of pCS. Total pCS levels were analyzed by HPLC using an Agilent Technologies 1100 liquid chromatograph with a quaternary pump, a diode array detector, a thermostatted column compartment, an autosampler, and an HP Compaq computer equipped with Agilent-

Chemstation software (Agilent Technologies, Santa Clara, CA, USA). Detailed information regarding serum sample preparation and HPLC analytical methodology for pCS assessment is depicted in Appendix B. The validation of the chromatographic method used is shown in Appendix C.

Urea, creatinine, uric acid, and phosphate were also measured by automated molecular absorption spectrometry methods with the C16000 Architect (Abbot Diagnostics, Abbott Park, IL, USA). The normal range is 15–50 mg/dL for urea, 0.60–1.20 mg/dL for creatinine, 3–7 mg/dL for uric acid, and 2.3–4.7 mg/dL and for phosphate. $\beta_2$-microglobulin concentrations were determined by a solid phase chemiluminescent immunoassay with the Siemens Inmulite 2500 Immunology Analyzer. The normal range is from 0.7 to 3.4 mg/L. Other laboratory measurements were performed using standard techniques at our hospital laboratory.

### 2.5. Statistical Analyses

Sample size calculation was estimated according to the expected different effect of the two dialyzers in the removal of pCS. Based on published data, the reduction rates of pCS by postdilution OL-HDF are around 40.0% with high-flux PS membranes [32]. Assuming a reduction rate difference of 15% at a standard deviation of 25%, and considering an error of beta = 0.8, a sample size of at least 29 patients was estimated. Finally, 35 patients were recruited.

Descriptive statistical results are presented as mean ± SD, median and interquartile range, and as a percentage of all patients as appropriate. For treatment comparisons of RR of each solute, linear mixed models were employed with covariate adjustment for the baseline level of the solute, treatment and sequence as fixed effects, and patient as a random term. The resulting *p* values were based on differences in least square means for the factor treatment and a significance level of 5% was employed. Comparison of other dialysis features (real length dialysis session, Qb, arterial pressure, venous pressure, initial and final body weight, ultrafiltration volume, volume of blood processed, convective volume, Kt, and Kt/V) between the two dialysis sessions was assessed using a paired sample *t*-test. Data collection of dialysis parameters was carried out using Nefrosoft software, version 7.0.1 (Visual-limes, Valencia, Spain). All analyses were conducted using R statistical software, version 4.0.3 (The R Foundation for Statistical Computing, Vienna, Austria), using the "lme4" and "RCommander" packages.

## 3. Results

### 3.1. Patient Characteristics

Thirty-five patients accepted to participate and were enrolled in the study. Patient characteristics are summarized in Table 2. All patients were Caucasian.

### 3.2. Dialysis Features

Most dialysis parameters were similar in both dialysis sessions, including duration, Qb, initial and final weight, ultrafiltration volume, arterial and venous pressures, and blood processed. Due to its higher permeability, the replacement fluid volume in postdilution OL-HDF was significantly greater with PS than that obtained with the PMMA dialyzer (Table 3). Both dialysis study sessions were performed without relevant clinical incidents (data not shown).

**Table 2.** Patients' characteristics.

| Characteristic | *n* = 35 |
|---|---|
| Age (yr) | 61.3 ± 15.4 |
| Sex (female/male; %) | 13/22 (37%/63%) |
| CKD etiology (*n*, %) | |
| -Nephrosclerosis | 6 (17%) |
| -Diabetic nephropathy | 7 (20%) |
| -Glomerular | 11 (31%) |
| -Interstitial | 2 (6%) |
| -Other causes | 7 (20%) |
| -Unknown | 2 (6%) |
| Dialysis vintage (mo; median, IQR) | 44 (24–162) |
| Comorbidity history (*n*, %) | |
| -Diabetes mellitus | 6 (18%) |
| -Coronary artery disease | 4 (12%) |
| -Chronic heart failure | 7 (21%) |
| -Cerebrovascular disease | 2 (6%) |
| -Peripheral vascular disease | 6 (18%) |
| Vascular access (*n*, %) | |
| -Arteriovenous fistula | 27 (77%) |
| -Arteriovenous graft | 1 (3%) |
| -Tunneled catheter | 7 (20%) |
| BMI (kg/m$^2$) | 25.4 ± 4.1 |
| nPNA (g/kg/d) | 1.26 ± 0.43 |
| Albumin (g/dL) | 3.6 ± 0.3 |
| Hemoglobin (g/dL) | 10.4 ± 1.2 |
| hs-CRP (mg/L) | 3.7 (1.0–8.2) |
| Total p-cresyl sulfate (mg/L) | 4.3 (1.4–7.6) |
| β2-microglobulin (mg/L) | 24.2 ± 11.5 |
| Creatinine (mg/dL) | 9.3 ± 2.6 |
| Uric acid (mg/dL) | 6.7 ± 1.2 |
| Phosphate (mg/dL) | 3.9 ± 1.0 |
| Calcium adjusted by albumin (mg/dL) | 9.0 ± 0.6 |
| iPTH (pg/mL) | 222 (144–340) |

Abbreviations: BMI, body mass index; CKD, chronic kidney disease; hs-CRP, high-sensitivity C-reactive protein; nPNA, normalized protein nitrogen appearance; iPTH, intact parathyroid hormone. Results are shown as mean ± s.d., median and interquartile range, or frequency (%), as appropriate.

**Table 3.** Comparison of dialysis features in the two study sessions.

| | BG-2.1U (PMMA) | TS-2.1 (PS) | *p* |
|---|---|---|---|
| Length session (min) | 247 ± 12 | 247 ± 12 | 0.9 |
| Qb (mL/min) | 380 ± 31 | 379 ± 30 | 0.9 |
| Initial weight (kg) | 72.2 ± 16.7 | 72.5 ± 16.2 | 0.1 |
| Final weight (kg) | 69.8 ± 16.4 | 70.0 ± 15.9 | 0.1 |
| Ultrafiltration (L) | 2.4 ± 1.1 | 2.5 ± 1.3 | 0.4 |
| Arterial pressure (mmHg) | −198 ± 34 | −187 ± 34 | 0.06 |
| Venous pressure (mmHg) | 157 ± 40 | 151 ± 38 | 0.07 |
| Blood processed (L) | 93.9 ± 9.7 | 93.8 ± 9.1 | 0.9 |
| Replacement volume (L) | 18.8 ± 2.8 | 30.3 ± 7.8 | <0.001 |

Abbreviations: Qb, blood flow rate; PMMA, polymethylmethacrylate; PS, polysulfone.

*3.3. Solute Reduction Ratios and Dialysis Dose*

The pCS RR with PMMA (BG-2.1U) membrane in OL-HDF was significantly higher than those obtained with PS (TS-2.1SL) membrane (Figure 1), with a relative difference of 28.6% (95% CI: 11.9 to 45.5; $p = 0.001$). Conversely, the β2-microglobulin RR value was higher with the PS membrane (Table 4), with a relative difference between dialyzers of 14.3% (95% CI: 12.6 to 16.0; $p < 0.001$). Except for phosphate, RRs for small molecules were

significantly lower with the PMMA membrane compared to the PS one, but with a mild effect size (Table 4). Relative differences between membranes in RRs were 1.8% (95% CI: 0.4 to 3.2; $p = 0.02$), 2.4% (95% CI: 1.1 to 3.7; $p < 0.001$), and 2.9% (95% CI: 1.6 to 4.1; $p < 0.001$), for urea, creatinine, and uric acid, respectively. No significant differences were found in phosphate RR, whereas the achieved dialysis doses estimated by ionic dialysance (Kt) and urea kinetics (Kt/V) were higher with PS membrane than those obtained with PMMA membrane (Table 4).

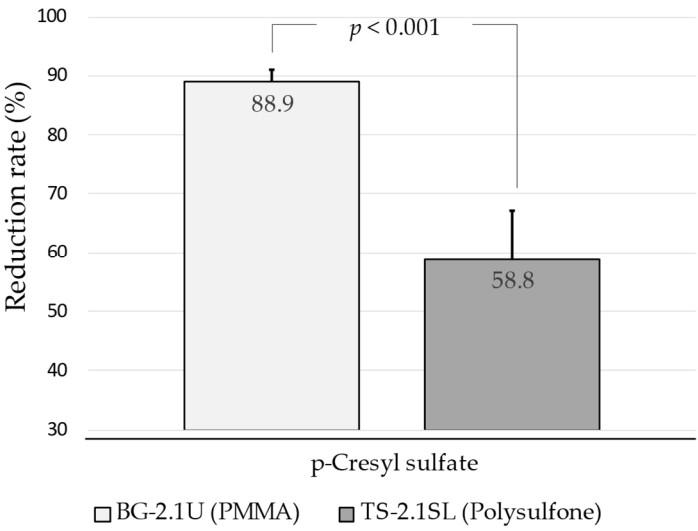

**Figure 1.** Comparison of p-cresyl sulfate reduction ratios (in percentages).

**Table 4.** Dialysis dose estimated by solute reduction ratios (in percentages), ionic dialysance (Kt), and urea kinetics (Kt/V).

| | BG-2.1U (PMMA) | TS-2.1SL (PS) | $p$ |
|---|---|---|---|
| Urea RR (60 Da) | $79.4 \pm 5.5$ | $81.1 \pm 5.4$ | 0.013 |
| Phosphate RR (95 Da) | $52.5 \pm 10.8$ | $54.6 \pm 13.5$ | 0.2 |
| Creatinine RR (113 Da) | $72.3 \pm 6.1$ | $74.6 \pm 5.7$ | <0.001 |
| Uric acid RR (168 Da) | $79.5 \pm 5.4$ | $82.1 \pm 4.4$ | <0.001 |
| β2-microglobulin RR (11,800 Da) | $67.5 \pm 7.1$ | $81.0 \pm 5.0$ | <0.001 |
| Kt (L) | $60.2 \pm 8.7$ | $65.5 \pm 9.4$ | 0.01 |
| Kt/V | $1.9 \pm 0.4$ | $2.0 \pm 0.5$ | 0.03 |

Abbreviations: PMMA, polymethylmethacrylate; PS, polysulfone; RR, reduction ratio.

## 4. Discussion

This is the first controlled study evaluating the effect of one of the last PMMA dialyzers suitable for HDF use, the BG-U membrane, on pCS removal by postdilution OL-HDF in prevalent HD patients. PBUT removal remains a challenge in the treatment of HD patients and strategies to decrease levels and hence toxicity, aiming to reduce the cardiovascular burden of these patients, are needed [23,33–36]. Whereas in the healthy kidney, PBUT clearance mostly depends on tubular secretion, in dialysis therapies the removal of these toxins is limited to the unbound fraction, not being affected by the pore size of the dialyzer [37], and only slightly by convective transport [38–40]. Conversely, PBUTs may be removed by using the adsorptive properties of certain biomaterials, including resins [41,42] and PMMA membranes [43]. Results suggest that BG-U dialyzers, compared to high-flux PS, are highly effective for reducing pCS levels. We also demonstrated that PMMA BG-U series achieved an acceptable convective volume for routine use, confirming a much higher permeability capacity than previous PMMA dialyzers.

With a slightly anionic PMMA membrane, the BG-U series were designed to offer higher biocompatibility to patients, with a controlled pore radius around 70 Å and a uniform distribution of pore size that guarantee high water permeability and porosity [44]. These modifications might enable the use of the BG-U dialyzers in OL-HDF with appropriate convective volume and acceptable albumin loss [25]. They combine the three mechanisms of diffusion, convection, and adsorption in a simple way, improving permeability and adsorption of not only low molecular weight proteins but also of higher molecular weight proteins up to 160,000 Da [45]. Although no other clinical study assessing the effect of BG-U dialyzers in pCS is available, we speculate that the higher efficacy on pCS removal observed in our study with BG-U series may be due to the adsorption properties of PMMA membranes [43]. Several studies have previously demonstrated the efficient removal of other PBUTs such as furancarboxylic acid and pentosidine [46,47], as well as inflammatory markers such as TNF-$\alpha$, IL-1$\beta$, IL-6, and C-reactive protein, by other PMMA dialyzers [48]. The high protein adsorption capacity of these membranes is due to their symmetrical pore structure which provides a large specific surface area [28,49]. Whereas in PMMA membranes the whole membrane thickness is involved in the separation process allowing toxin adsorption, in PS membranes (with asymmetrical pores) only a fine layer of $\approx$1 μm is responsible for the separation process, while the remaining membrane thickness has structural functions only [20]. These differences in membrane structure may explain the distinct pCS removal profile obtained with the two tested dialyzers, which constitutes the most original finding of this study. Although the results of short-term studies such as this one may not adequately reflect long-term trends and patient outcomes, we think that the reduction of pCS observed with the PMMA BG-U dialyzer could be clinically relevant. The elevated cardiovascular morbidity and mortality risk of HD patients has been repeatedly associated with levels of pCS and other PBUTs [33–36]. Moreover, recent research suggests that these PBUTs accelerate the progression of CV disease, bone disorders, and neurological complications among CKD patients [50,51]. Results from the annual survey of the Japanese Nationwide Dialysis Registry suggest that the use of PMMA membranes may reduce mortality in HD patients [52]. However, further long-term prospective studies are needed to clarify these findings.

In parallel, we also observed that PMMA BG-U series may be appropriately used by postdilution HDF, confirming the higher permeability capacity observed with new designed PMMA dialyzers [24]. The mean convective volume obtained in our study was close to the 21 L threshold which has been associated with better survival in large randomized clinical trials [53]. However, compared with PS dialyzer, which reached higher replacement volume, the convective efficacy estimated by β2-microglobulin RRs was 14 percentage points lower. These differences are comparable to those recently obtained by Maduell et al. in a safety and efficacy evaluation of PMMA NF-U series [24]. This latest generation of PMMA dialyzers may allow the achievement of high convective volume with no significant albumin loss. All these data suggest that the indication of new high-flux PMMA dialyzers in postdilution HDF may represent a practical compromise between efficient convective and adsorptive dialysis treatment.

### 4.1. Strengths and Limitations

Strengths of this study were its cross-sectional design, with each patient as their own control, and the use of a chromatographic method for the assessment of pCS levels, which was validated according to the European Medicines Agency (EMA) and the Federal Drug Administration (FDA) [54,55]. To eliminate confounding, the same dialysis features were applied to both HD sessions. There are additional limitations, starting with its short-term nature plus the relatively small sample size that leads us to consider this as a pilot trial in need of verification. We did not collect the dialysis fluid to quantify the elimination of toxins. We also did not assess the albumin loss in dialysate. However, available data with BG-U series suggest this PMMA dialyzer as highly adsorptive but with the same cut-off as PS dialyzers [25,56,57], and consequently, its indication in HDF seems safe and

appropriate. Moreover, albumin loss is only one of many factors contributing to the risk of hypoalbuminemia in dialysis patients [58]. Additionally, strategies for reducing the risk of malnutrition in this population include improving systemic inflammation by increasing uremic toxin removal and optimizing the biocompatibility of the dialysis procedure [59]. Although these factors may be further improved by new PMMA dialyzers, we acknowledge the lack of information on albumin dialysate loss in our study, which makes our previous statement speculative.

*4.2. Conclusion and Clinical Implications*

This study suggests that OL-HDF with PMMA BG-U series is highly effective for the removal of pCS, enabling an acceptable clearance of β2-microglobulin and small solutes. These results support the continuing use of hydrophobic and cationic adsorptive PMMA membranes as a good alternative in HD treatment, which could potentially enhance the clinical benefits in patients on renal replacement therapies. With an increasing number of dialyzer options, there is a need to further examine the clinical effects of removal of PBUTs on quality of life and survival in HD patients, whose life expectancy continues to be unacceptably low.

**Author Contributions:** Research idea and study design: P.M., J.P., B.V., M.G.-M.; data acquisition: P.M., J.P., C.E., M.A.M.-G., B.V., M.G., M.G.-V.; data analysis/interpretation: P.M., J.P., M.A.M.-G., B.V., M.G.-M.; statistical analysis: P.M., M.D.M.; supervision and mentorship: F.M. All authors have contributed important intellectual content during manuscript drafting or revision and accept accountability for the overall work. All authors have read and agreed to the published version of the manuscript.

**Funding:** This study was partially funded by Palex through the acquisition of all the chemicals and reagents necessary to carry out the synthesis of pCS and the solid phase extraction cartridges required to determine pCS in all serum samples by HPLC.

**Institutional Review Board Statement:** The study was conducted according to the guidelines of the Declaration of Helsinki and approved by the Ethics Committee of Hospital Universitari Dr. Peset (protocol code 14/012).

**Informed Consent Statement:** Informed consent was obtained from all subjects involved in the study.

**Data Availability Statement:** Not applicable.

**Acknowledgments:** The authors wish to thank all patients, but also volunteers, laboratory technicians, and the nurses and physicians taking care of both patients and samples, of the Departments of Clinical Analysis, Pharmacy, and Nephrology at Hospital Universitari Dr. Peset in Valencia. The authors would also like to specifically thank Francisco Estevan of the Chemistry Faculty at the University of Valencia for his selfless collaboration in the synthesis and characterization by 1H-RMN, 13C-RMN, and mass spectrometry of the pCS. We are also grateful to Manuel Fraile (CNC) for his stimulating comments and guidance in the study. We thank Marta Maojo for professional translating assistance.

**Conflicts of Interest:** The results presented in this paper have not been published previously in whole or part, except in abstract format. P.M. acknowledges consultation or speaker fees from Palex and Medtronic. F.M. acknowledges consultancy and lectures fees from Baxter, Fresenius Medical Care, Medtronic, and Nipro. The authors declare no other conflicts of interest that might be perceived as affecting the objectivity of this study. The funders had no role in the design of the study; in the collection, analyses, or interpretation of data; in the writing of the manuscript, or in the decision to publish the results.

**Appendix A. Study Investigators**

The following investigators participated in the study: Pablo Molina, Julio Peiró, María A. Martínez-Gómez, Belén Vizcaíno, Cristina Esteller, Mercedes González-Moya, María García-Valdelvira, Josep Ventura, Erika Pérez-Zafra, Emma Calatayud, María Montesa,

Alejandro Valero, Sandra Beltrán, Julia Kanter, Nuria Estañ, Mónica Climente, Asunción Sancho, Mariola D Molina, Francisco Maduell.

**Appendix B. Measurement of pCS Levels**

*Appendix B.1. Chemicals and Reagents for pCS Measurement*

The pCS needed to develop an HPLC method was synthetized in the Inorganic Chemistry Department of the Faculty of Chemistry of the University of Valencia, as described previously by Feigenbaum and Neuberg [60]. In this procedure, p-cresol dissolved in pyridine is converted to pCS by the slow addition drop by drop of chlorosulfonic acid. pCS is crystallized out of the reaction mixture by addition of ethanol and finally purified by recrystallization with water. The obtained product was lyophilized to eliminate the excess of water derived from the recrystallization process. The identity and purity of the synthetized compound was confirmed by mass spectrometry (Figure A1), proton nuclear magnetic resonance ($^1$H-RMN, Figure A2), and $^{13}$C-RMN (Figure A3).

Methanol and formic acid employed for the development of the HPLC method were purchased from Scharlab S.L. (Spain) and Fluka Analytical (Sweden), respectively. Deionized water was used to prepare the mobile phase used in the chromatographic analysis. P-cresol, pyridine, ethanol, and chlorosulfonic acid necessary for the organic synthesis of pCS were purchased from Sigma-Aldrich (Spain). To develop the chromatographic method, free serum was collected from healthy volunteers at Doctor Peset University Hospital (Valencia, Spain) and kept frozen at $-20\,^\circ$C until analysis.

*Appendix B.2. Preparation of pCS Standard Solutions*

Stock I solution of pCS was prepared by dissolving 12 mg of pCS in 2 mL of water and stored at 4 $^\circ$C. Stock II solution of pCS was prepared by diluting 167 µL of Stock I solution with 833 µL of water. For the calibration curve, seven standard solutions of pCS were prepared by making serial dilutions from stock solutions with free serum over the range 0.05–6.25 mg/mL.

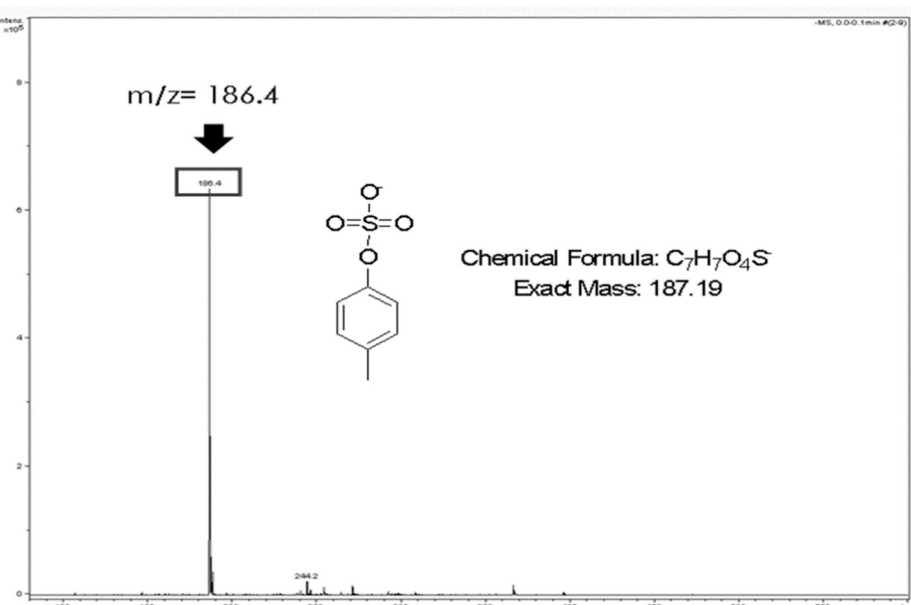

**Figure A1.** pCS mass spectrum.

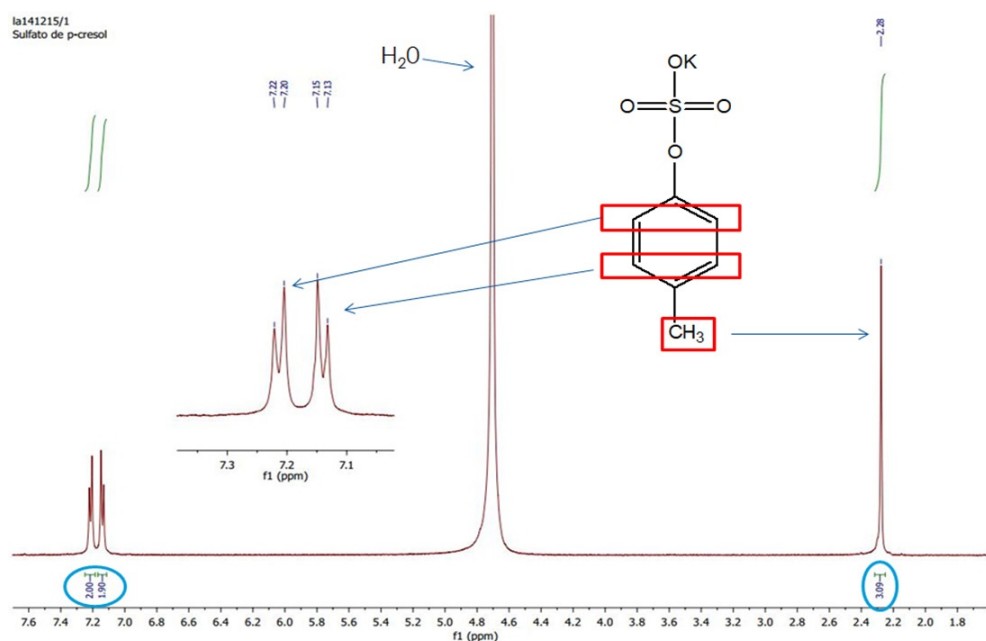

**Figure A2.** Proton nuclear magnetic resonance (¹H-RMN) pCS spectrum.

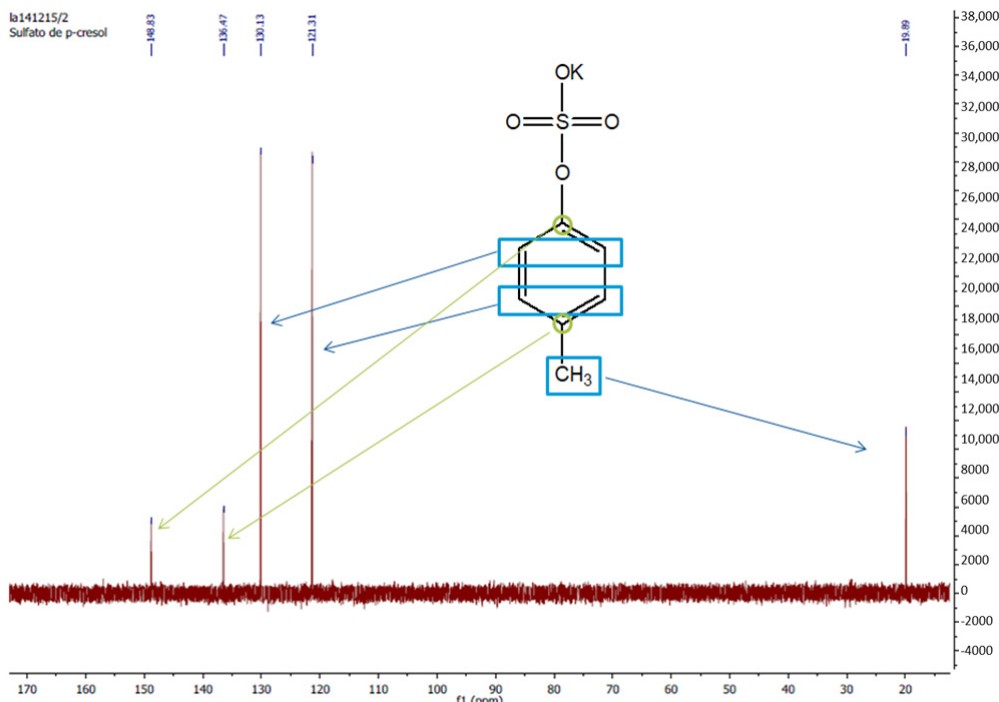

**Figure A3.** Carbon nuclear magnetic resonance (¹³C-RMN) pCS spectrum.

*Appendix B.3. Serum Sample Preparation*

Methanol deproteinization of blood samples was used to avoid hydrolysis of pCS by acids [61]. Serum samples and standard solutions were treated equally. In all cases, an aliquot of 500 μL was added to 1 mL of methanol and was then incubated at room temperature for 20 min. After that, the mixture was centrifugated for 10 min at 3500 rpm and the supernatant was collected. Cyano bonding cartridges (Discovery® DSC-CN SPE Tube, bed weight 500 mg, volume 3 mL; Supelco, Bellefonte, PA, USA) were used for solid phase extraction. Before the extraction, cartridges were conditioned with 2 mL of methanol, centrifuged subsequently for 1 min at 3500 rpm, and followed by the addition

of 2 mL of water and centrifuged 1 min at 3500 rpm. After the conditioning procedure, supernatant of the deproteinized samples or standard solutions was loaded into a cartridge and was centrifuged for 1 min at 3500 rpm. The cartridge was put inside a clean borosilicate tube with 1 mL of methanol. After being centrifuged for 1 min at 3500 rpm, the eluted fraction was collected and was evaporated at 37 °C under a vacuum of 600 mm Hg for 60 min in a Heidolph Synthesis 1 Multi-evaporator (Heidolph Instruments GmbH & Co.KG, Schwabach, Germany). Dry residue was reconstituted with 150 μL of mobile phase and was finally transferred into an HPLC vial for analysis.

*Appendix B.4. HPLC Analytical Methodology*

pCS was analyzed by HPLC using an Agilent Technologies 1100 liquid chromatograph with a quaternary pump, a diode array detector, a thermostatted column compartment, an autosampler, and an HP Compaq computer equipped with Agilent-Chemstation software (Agilent Technologies, Santa Clara, CA, USA). The chromatographic separations were performed on a Kromasil® RP C18 analytical column (150 mm length × 4.6 mm i.d., 5 μm particle diameter; Análisis Vínicos, Spain). The samples (20 μL each) were injected through a Rheodyne valve (Rheodyne, Cotati, CA, USA). The flow rate was set to 1 mL/min, temperature to 25 °C, and fluorescence detection with 214 nm for excitation and 306 nm for emission and detection [60]. Mobile phase was composed of 50 mM formic acid and methanol. An elution gradient was necessary: t = 0 min, formic acid/methanol (65:35, *v/v*); t = 15 min, formic acid/methanol (25:75, *v/v*); t = 19 min, formic acid/methanol (65:35, *v/v*). The column was equilibrated for 30 min prior to injection of samples. The peak area of pCS was measured in each chromatogram. Retention time of pCS was 13 min.

Formic acid and methanol solutions were vacuum filtered through 0.45 μm nylon membranes (Micron Separations, Westboro, MA, USA) and sonicated prior to HPLC analysis. An SC2 analytical microbalance (Sartorius Mechatronics, S.A., Madrid, Spain) was used to weigh pCS.

*Appendix B.5. Chromatographic Method Validation*

The chromatographic method was validated according to the EMA [54] and FDA [55]. For each drug, linearity, accuracy, repeatability, intermediate precision, recovery, specificity, limit of detection and quantification, and system suitability were evaluated [62].

Linearity was demonstrated by analyzing the pCS standard solutions over the range 0.05–6.25 mg/mL; a calibration curve was performed by plotting peak area against drug concentration; the coefficient of determination (r2) was calculated. The selected concentrations covered the range of expected pCS serum concentrations in patients on dialysis, according to [63] and to our preliminary studies. Accuracy was determined by comparing mean estimated concentration with the nominal value at four pCS concentration levels (0.05, 0.52, 2.60, and 6.25 mg/mL). Relative errors (REs) were also calculated. Repeatability (intra-day assay precision) was determined by analyzing four pCS standards (0.05, 0.52, 2.60, and 6.25 mg/mL) twice and calculating the RSD for each concentration level. Intermediate precision (inter-day assay precision) was determined by analyzing four pCS standards (0.05, 0.52, 2.60, and 6.25 mg/mL) daily for two days and calculating the RSD for each concentration level. Specificity of the method was ascertained by evaluating the presence of interferences at the retention time of pCS. Limit of detection (LOD) and limit of quantification (LOQ): LOD and LOQ were calculated using the following equations: LOD = 3·σ/S and LOQ = 10·σ/S; where σ is the standard deviation of y-intercepts of regression lines and S is the slope of the calibration curve. System suitability specifications and tests (SSTs) were determined from ten replicate injections of pCS standard solutions of 0.05, 0.52, 2.60, and 6.25 mg/mL. Theoretical plates (N), tailing factor (T), resolution (Rs), and repeatability (RSD of retention time and area) were determined as the mean of the ten values obtained for each parameter.

**Appendix C. Chromatographic Method Validation**

In the experimental conditions selected, the analytical performance parameters suggested by ICH guidelines were evaluated:

- Linearity: r2 was 0.9999. The slope but not the intercept was statistically significant at 95% confidence level, the equation being y = 165.08·x.
- Accuracy: The method was found to be accurate since percentage of recovery was between 98.9 and 100.6%.
- Precision: The % RSD was ≤1.2% and ≤0.8% in all cases for repeatability and intermediate precision, respectively, which indicated that the method was precise.
- Specificity: It was adequate since no interferences were observed at retention time of pCS.
- LOD and LOQ: Under the experimental conditions used, LOD and LOQ were 0.009 mg/mL and 0.029 mg/mL, respectively.

About SST parameters, all values were within the acceptance criteria: N ≥ 2000, T ≤ 2, Rs > 2, and RSD ≤ 1%. In conclusion, all the criteria were acceptable according to ICH guidelines, and the proposed HPLC method was adequate to determine pCS in serum samples.

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
