# Peer review of "Combining Diffusion, Convection and Absorption: A Pilot Study of Polymethylmethacrylate versus Polysulfone Membranes in the Removal of P-Cresyl Sulfate by Postdilution On-Line Hemodiafiltration"

_kidneydial, doi:10.3390/kidneydial1020015_

Round 1
Reviewer 1 Report
To test the hypothesis that high-flux polymethyl methacrylate (PMMA) dialysis membranes with adsorptive capacity increase p-cresyl sulfate (pCS) removal compared to polysulfone (PS) membranes, in a post-dilution on-line hemodiafiltration (OL-HDF) session the authors conducted the present study in 35 stable hemodialysis patients. The reduction ratios (RRs) for pCS were higher with the PMMA membrane compared with polysulfone membrane, whereas the β2-microglobulin RRs, Kt, Kt/V, and the convection volume were significantly higher with polysulfone membrane. According to these findings, the authors concluded that pCS removal by OL-HDF was superior with high-flux PMMA membranes, appearing to be a good dialysis strategy for improving dialytic clearance of pCS.
Although the theme of this study is intriguing, there are some concerns to be addressed.
1. The authors concluded that pCS removal by OL-HDF was superior with high-flux PMMA membranes, appearing to be a good dialysis strategy for improving dialytic clearance of pCS, enabling an adequate β2-microglobulin and small solute clearances, although the clearance of β2-microglobulin by PMMA membrane were significantly lower than that by PS membrane. I wonder why the authors consider adequate for β2-microglobulin clearance of 67.5%.
2. The discussion is described like a narrative review. It should be more concise.
3. The data in Figure 1 are duplicates of a part of the data in Table 4. Duplicate data should be shown with one of the two, not both.
4. The unit of values should be shown in Table 4.
5. There are many spelling errors to be corrected.
Page 1, Abstract: polymethylmetacriylate → polymethylmethacrylate
Page 2, line 20: polysulphone → polysulfone
Page 3, Table 1: Inuline → Inulin
Page 4, line 17: thermostated → thermostatted
Page 4, line 42: vesion → version
Page 5, Table 2: Tunnelled catheter → Tunneled catheter; Albumin (g/L) → Albumin (g/dL); Hs-CRP (mg/dL) → Hs-CRP (mg/L)
Page 8, line 34: R+cent → Recent
Page 8, line 45: 14% → 14 percentage point
Page 10, lines 12 and 13: recristallinzation → recrystallization
Page 10, line 4 from the bottom: solucion → solution
Page 12, line 10 from the bottom: thermostated → thermostatted
Author Response
- Comment 1. The authors concluded that pCS removal by OL-HDF was superior with high-flux PMMA membranes, appearing to be a good dialysis strategy for improving dialytic clearance of pCS, enabling an adequate β2-microglobulin and small solute clearances, although the clearance of β2-microglobulin by PMMA membrane were significantly lower than that by PS membrane. I wonder why the authors consider adequate for β2-microglobulin clearance of 67.5%.
Thank you for pointing this out. We have toned down the sentence, considering the 67.5% reduction ratio as barely acceptable. Both the abstract and the discussion have been modified accordingly.
- Comment 2. The discussion is described like a narrative review. It should be more concise.
We appreciate your point. In this new version of the manuscript, we have revised the Discussion section to make it more concise. Several paragraphs have been modified and shortened.
- Comment 3. The data in Figure 1 are duplicates of a part of the data in Table 4. Duplicate data should be shown with one of the two, not both.
Thank you for the comment. Duplicated data has been deleted from Figure 1 and Table 4, for avoiding overlapping.
- Comment 4. The unit of values should be shown in Table 4.
Thank you for pointing this out. The unit values have been added to the table.
- Comment 5. There are many spelling errors to be corrected.
- Page 1, Abstract: polymethylmetacriylate → polymethylmethacrylate
- Page 2, line 20: polysulphone → polysulfone
- Page 3, Table 1: Inuline → Inulin
- Page 4, line 17: thermostated → thermostatted
- Page 4, line 42: vesion → version
- Page 5, Table 2: Tunnelled catheter → Tunneled catheter; Albumin (g/L) → Albumin (g/dL); Hs-CRP (mg/dL) → Hs-CRP (mg/L)
- Page 8, line 34: R+cent → Recent
- Page 8, line 45: 14% → 14 percentage point
- Page 10, lines 12 and 13: recristallinzation → recrystallization
- Page 10, line 4 from the bottom: solucion → solution
- Page 12, line 10 from the bottom: thermostated → thermostatted
Thank you for picking up on this, we have corrected all the spelling errors.
Reviewer 2 Report
The manuscript entitled "Combining Diffusion, Convection and Absorption: A Pilot Study of Polymethylmetacriylate versus Polysulfone Membranes in the Removal of P-Cresyl Sulfate by Postdilution On-Line Hemodiafiltration" is well designed, performed and presented. This study shows that the use of this new membrane could potentially enhance, hopefully not marginally, the clinical benefits in patients on renal replacement therapies. Limitations of the study are honestly listed. Generally, it is a well written and well-organized study.
Author Response
Thank you.
Reviewer 3 Report
The study is well conducted. The analysis is valid. The manuscript is clearly written. I have few comments to be considered for clarity.
Please provide detail of cross-over process. Did you randomly assign patients to receive PMMA or PS first? What is the wash out period? If so, please provide patient characteristics between patients who were assigned to receive PMMA and TS first in table 2
How did you derive the sample size of 35 patients? Did you do sample size calculation
Author Response
- Comment 1. The study is well conducted. The analysis is valid. The manuscript is clearly written. I have few comments to be considered for clarity.
We are pleased that you thought the study was well conducted and written.
- Comment 2. Please provide detail of cross-over process. Did you randomly assign patients to receive PMMA or PS first? What is the wash out period? If so, please provide patient characteristics between patients who were assigned to receive PMMA and TS first in table 2.
The information about the allocation process has been expanded. Patients were randomly assigned to receive PMMA or PS first. During the 4-week wash-out period, patients remained in their usual HD treatment plan with no changes; all received post-dilution OL-HDF with high-flux, PS FX-100 (Fresenius Medical Careâ). Due to the cross-sectional design of the study, with each patient as their own control, there is no need for a new table.
- Comment 3. How did you derive the sample size of 35 patients? Did you do sample size calculation?
The authors appreciate and agree with this comment. The sample size calculation was estimated according to the expected different effect of the two dialyzers in the removal of pCS. Accordingly, we have added the following text to the Methods section, along with a new citation:
“Sample size calculation was estimated according to the expected different effect of the two dialyzers in the removal of pCS. Based on published data, the reduction rates of pCS by postdilution OL-HDF is around 40.0 % with high-flux PS membranes (Meert;2009). Assuming a reduction rate difference of 15% at a standard deviation of 25%, and considering an error of beta=0.8, a sample size of at least 29 patients was estimated. Finally, 35 patients were recruited.”
Meert N, Eloot S, Waterloos MA, Van Landschoot M, Dhondt A, Glorieux G, Ledebo I, Vanholder R. Effective removal of protein-bound uraemic solutes by different convective strategies: a prospective trial. Nephrol Dial Transplant. 2009 Feb;24(2):562-70. doi: 10.1093/ndt/gfn522. Epub 2008 Sep 22. PMID: 18809977.
Round 2
Reviewer 1 Report
I have no further comments.
Reviewer 3 Report
All of my comments have been addressed.